# Synthetic Benchmarks for Scientific Research in Explainable Machine Learning

**Yang Liu**[*]
Abacus.AI
San Francisco, CA 94103
`yang@abacus.ai`

**Sujay Khandagale**[*]
Abacus.AI
San Francisco, CA 94103
`sujay@abacus.ai`

**Colin White**
Abacus.AI
San Francisco, CA 94103
`colin@abacus.ai`

**Willie Neiswanger**
Stanford University
Stanford, CA 94305
`neiswanger@cs.stanford.edu`

## Abstract

As machine learning models grow more complex and their applications become more high-stakes, tools for explaining model predictions have become increasingly important. This has spurred a flurry of research in model explainability and has given rise to feature attribution methods such as LIME and SHAP. Despite their widespread use, evaluating and comparing different feature attribution methods remains challenging: evaluations ideally require human studies, and empirical evaluation metrics are often data-intensive or computationally prohibitive on real-world datasets. In this work, we address this issue by releasing XAI-BENCH: a suite of synthetic datasets along with a library for benchmarking feature attribution algorithms. Unlike real-world datasets, synthetic datasets allow the efficient computation of conditional expected values that are needed to evaluate ground-truth Shapley values and other metrics. The synthetic datasets we release offer a wide variety of parameters that can be configured to simulate real-world data. We demonstrate the power of our library by benchmarking popular explainability techniques across several evaluation metrics and across a variety of settings. The versatility and efficiency of our library will help researchers bring their explainability methods from development to deployment. Our code is available at `https://github.com/abacusai/xai-bench`.

## 1 Introduction

The last decade has seen a huge increase in applications of machine learning in a wide variety of high-stakes domains, such as credit scoring, fraud detection, criminal recidivism, and loan repayment [35, 10, 36, 9]. With the widespread deployment of machine learning models in applications that impact human lives, research on model explainability is becoming increasingly more important. The applications of model explainability include debugging, legal obligations to give explanations, recognizing and mitigating bias, data labeling, and faster adoption of machine learning technologies [33, 55, 7, 18]. Many different methods for explainability are actively being explored including logic rules [22, 50, 44], hidden semantics [54], feature attribution [40, 33, 39, 14, 48], and explanation by example [31, 12]. The most common type of explainers are post-hoc, local feature attribution methods [55, 33, 1, 40, 39, 14], which output a set of weights corresponding to the importance of each feature for a given datapoint and model prediction. Although various feature attribution methods are being deployed in different use cases today, currently there are no widely adopted methods to easily

---

[*]Equal contribution

*evaluate and/or compare* different feature attribution algorithms. Indeed, evaluating the effectiveness of explanations is an intrinsically human-centric task that ideally requires human studies. However, it is often desirable to develop new explainability techniques using empirical evaluation metrics before the human trial stage. Although empirical evaluation metrics have been proposed, many of these metrics are either computationally prohibitive or require strong assumptions, to compute on real-world datasets. For example, a popular method for feature attribution is to approximate Shapley values [33, 16, 32, 48], but computing the distance to ground-truth Shapley values requires estimating exponentially many conditional feature distributions, which is not possible to compute unless the dataset contains sufficiently many datapoints across exponentially many combinations of features.

In this work, we overcome these challenges by releasing a suite of synthetic datasets, which make it possible to efficiently benchmark feature attribution methods. The use of synthetic datasets, for which the ground-truth distribution of data is known, makes it possible to exactly compute the conditional distribution over any set of features, thus enabling computations of many feature attribution evaluation metrics such as distance to ground-truth Shapley values [33], remove-and-retrain (ROAR) [26], faithfulness [3], and monotonicity [34]. Our synthetic datasets offer a wide variety of parameters which can be configured to simulate real-world data and have the potential to identify subtle failures, such as the deterioration of performance on datasets with high feature correlation. We give examples of how real datasets can be converted to similar synthetic datasets, thereby allowing explainability methods to be benchmarked on realistic synthetic datasets.

We showcase the power of our library by benchmarking popular explainers such as SHAP [33], LIME [40], MAPLE [39], SHAPR [1], L2X[14], and breakDown [47], with respect to a broad set of evaluation metrics, across a variety of axes of comparison, such as feature correlation, model type, and data distribution type. Our library is designed to substantially accelerate the time it takes for researchers and practitioners to move their explainability algorithms from development to deployment. All of our code, API docs, and walkthroughs are available at `https://github.com/abacusai/xai-bench`. We welcome contributions and hope to grow the repository to handle a wide variety of use-cases. We expect the scope and breadth of our framework to increase over time.

**Our contributions.** We summarize our main contributions below.

- We release a set of synthetic datasets with known ground-truth distributions, along with a library that makes it possible to efficiently evaluate feature attribution techniques with respect to ten different metrics. Our synthetic datasets offer a wide variety of parameters that can be configured to simulate real-world applications.
- We demonstrate the power of our library by benchmarking popular explainers such as SHAP [33], LIME [40], MAPLE [39], SHAPR [1], L2X[14], and breakDown [47].

## 2   Related Work

Model explainability in machine learning has seen a wide range of approaches, and multiple taxonomies have been proposed to classify the different types of approaches. Zhang et al. [55] describe three dimensions of explainability techniques: passive/active, type of explanation, and local/global explainations. The types of explanations they identified are logic rules [22, 50, 44], hidden semantics [54], feature attribution [40, 33, 39, 14, 48, 1], and explanation by example [31, 12]. Other surveys on explainable AI include Arrieta et al. [6], Adadi and Berrada [2], and Došilović et al. [20].

Techniques for feature attribution include approximating Shapley values [33, 16, 32, 48], approximating the model locally with a more explainable model [40], and approximating the mutual information of each feature with the label [14]. In Appendix E, we give descriptions and implementation details of the feature attribution methods we implemented. Other work has also identified failure modes for some explanation techniques. For example, recent work has shown that explanation techniques are susceptible to adversarial feature perturbations [19, 46, 24], high feature correlations [29], and small changes in hyperparameters [23, 8].

### 2.1   Benchmarking Explainability Techniques

One recent work [27] gave an experimental survey of explainability methods, testing SHAP [33], LIME [40], Anchors [41], Saliency Maps [45], and Grad-CAM++ [11], and their proposed Ex-Matchina on image, text, audio, and sensory datasets. They use human labeling via Mechanical Turk as an evaluation metric. Another work [7] gave an experimental survey of several algorithms includ-

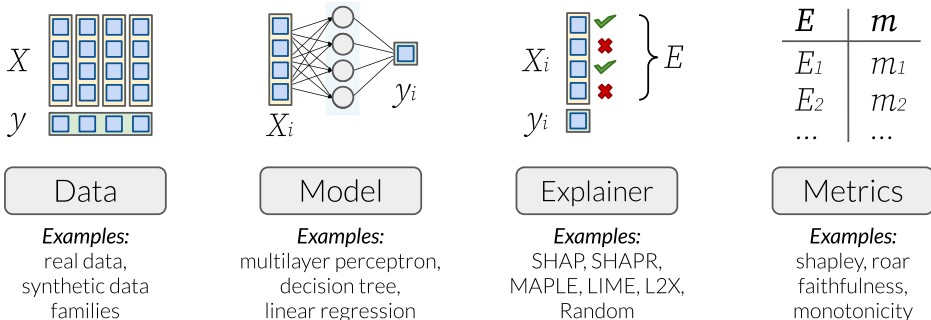

Figure 1: Overview of the main components in XAI-BENCH.

ing local/global, white-box/black-box, and supervised/unsupervised techniques. The only feature attribution algorithms they tested were SHAP and LIME. Another recent work gives a benchmark on explainability for time-series classification [21]. Another recent work [18] gives a set of benchmark natural language processing (NLP) datasets aimed at comparing explainability methods. This work releases multiple datsets with human-annotated explanations, as well as a few newly proposed metrics specifically chosen to capture the explainability of predictions in NLP applications. Finally, concurrent work [4] releases a library with several evaluation metrics for local linear explanation methods and uses the library to compare LIME and SHAP. To the best of our knowledge, no prior work has released a library with ten different evaluation metrics or released a set of synthetic datasets for explainability with more than one tunable parameter.

## 2.2 Explainability evaluation metrics

While the "correctness" of feature attribution methods may be subjective, comparisons between methods are often based on human studies [28, 42, 43]. However, human studies are not always possible, and several empirical (non-human) evaluation metrics have been proposed. Faithfulness [3] measures the correlation between the weights of the feature attribution algorithm, and the effect of the features on the performance of the model. Monotonicity [34] checks whether iteratively adding features from least weighted feature to most weighted feature, causes the prediction to monotonically improve. By retraining a model with subsets of features ablated, ROAR [26] uses a new model with partially ablated input features to evaluate a feature attribution technique while avoiding problems with distribution shift. Note that all of the above metrics evaluate feature importance by computing the effect of removing the feature from a single set of features $S$. In contrast, Shapley values [33, 16, 32, 48] evaluate all possible sets $S$ that a feature can be removed from to compute an average effect.

# 3 Evaluation Metrics

## 3.1 Preliminaries

Now we give definitions and background information used throughout the next three sections. Given a distribution $\mathcal{D}$, each datapoint is of the form $(\boldsymbol{x}, y) \sim \mathcal{D}$, where $\boldsymbol{x}$ denotes the set of features, and $y$ denotes the label. We assume that $\boldsymbol{x} \in [0, 1]^D$ and $y \in [0, 1]$, yet all of the concepts we discuss can be generalized to arbitrary categorical and real-valued feature distributions and labels. Assume we have a training set $\mathcal{D}_{\text{train}}$ and a test set $\mathcal{D}_{\text{test}}$, both drawn from $\mathcal{D}$. We train a model $f : [0, 1]^D \to [0, 1]$ on the training set. Common choices for $f$ include a neural network or a decision tree.

A *feature attribution method* is a function $g$ which can be used to estimate the importance of each feature in making a prediction. That is, given a model $f$ and a datapoint $\boldsymbol{x}$, then $g(\boldsymbol{x}, f) = \boldsymbol{w} \in [-1, 1]^D$, where each output weight $w_i$ corresponds to the relative importance of feature $i$ when making the prediction $f(\boldsymbol{x})$. Common choices for $g$ include SHAP [33] or LIME [40].

## 3.2 Metrics

In this section, we define several different evaluation metrics for explainability methods. Each evaluation metric has pros and cons, and the most useful metric depends on the situation. We give more details of the pros and cons in Section 3.3. A *feature attribution evaluation metric* is a function which evaluates the weights of a feature attribution method on a datapoint $\boldsymbol{x}$. For example, given

a datapoint $\boldsymbol{x}$ and a set of feature weights $\boldsymbol{w} = g(\boldsymbol{x}, f)$, then a value near zero indicates that $g$ did not provide an accurate feature attribution estimate for $\boldsymbol{x}$, while a value near one indicates that $g$ did provide an accurate feature attribution estimate.

Many evaluation metrics involve evaluating the change in performance of the model when a subset of features of a datapoint are removed. In order to measure the true marginal improvement for a set of features $S$, we evaluate the model when replacing the features $S$ with their expected values conditioned on the remaining features. Formally, given a datapoint $\boldsymbol{x} \sim \mathcal{D}$ and a set of indices $S \subseteq \{1, \cdots, D\}$, we define $\mathcal{D}(\boldsymbol{x}_S)$ as the conditional probability distribution $\boldsymbol{x}' \sim \mathcal{D}$ such that $x_i' = x_i$ for all $i \in S$. In other words, given $\boldsymbol{x}$ and $S$, we have

$$p(\boldsymbol{x}' \sim \mathcal{D}(\boldsymbol{x}_S)) = p(\boldsymbol{x}' \sim \mathcal{D} \mid x_i' = x_i \text{ for all } i \in S). \tag{1}$$

By this definition, $\mathcal{D}(\boldsymbol{x}_\emptyset) = \mathcal{D}$, and if we define $F = \{1, \cdots, D\}$, then $\boldsymbol{x}' \sim \mathcal{D}(\boldsymbol{x}_F)$ is equal to $\boldsymbol{x}$ with probability 1. Given a datapoint $\boldsymbol{x}$, a model $f$, and a weight vector $\boldsymbol{w}$, the first evaluation metric, **faithfulness** (faith$-$) [3], is defined as follows:

$$\text{faith}- = \text{Pearson}\left(\left|\mathbb{E}_{\boldsymbol{x}' \sim \mathcal{D}(\boldsymbol{x}_{F \setminus i})}[f(\boldsymbol{x}')] - f(\boldsymbol{x})\right|_{1 \leq i \leq D}, [w_i]_{1 \leq i \leq D}\right). \tag{2}$$

Intuitively, faith$-$ computes the Pearson correlation coefficient [52] between the weight vector $\boldsymbol{w}$ and the approximate marginal contribution $\left|\mathbb{E}_{\boldsymbol{x}' \sim \mathcal{D}(\boldsymbol{x}_{F \setminus i})}[f(\boldsymbol{x}')] - f(\boldsymbol{x})\right|$ for each feature $i$. We also study a new variant of faithfulness: instead of computing the marginal improvement between the original datapoint, $\boldsymbol{x}$, and the datapoint with one feature replaced, $\boldsymbol{x}_{F \setminus i}$, we instead compute the marginal improvement between the global mean datapoint, $\boldsymbol{x}_\emptyset$, and the datapoint with one feature *added*, $\boldsymbol{x}_{\{i\}}$. Formally,

$$\text{faith}+ = \text{Pearson}\left(\left|\mathbb{E}_{\boldsymbol{x}' \sim \mathcal{D}(\boldsymbol{x}_{\{i\}})}[f(\boldsymbol{x}')] - \mathbb{E}_{\boldsymbol{x}' \sim \mathcal{D}(\boldsymbol{x}_\emptyset)}[f(\boldsymbol{x}')]\right|_{1 \leq i \leq D}, [w_i]_{1 \leq i \leq D}\right). \tag{3}$$

The next metric computes the marginal improvement of each feature ordered by the weight vector $\boldsymbol{w}$ *without replacement*, and then computes the fraction of indices $i$ such that the marginal improvement for feature $i$ is greater than the marginal improvement for feature $i+1$. Formally, define $S^-(\boldsymbol{w}, i)$ as the set of $i$ least important weights, define $S^+(\boldsymbol{w}, i)$ as the set of $i$ most important weights, and let $S^-(\boldsymbol{w}, 0) = \emptyset$. Given a datapoint $\boldsymbol{x}$, a model $f$, and a weight vector $\boldsymbol{w}$, we define **monotonicity** (mono$-$) [34] as follows:

$$\delta_i^- = \mathbb{E}_{\boldsymbol{x}' \sim \mathcal{D}(\boldsymbol{x}_{S^-(\boldsymbol{w}, i+1)})}[f(\boldsymbol{x}')] - \mathbb{E}_{\boldsymbol{x}' \sim \mathcal{D}(\boldsymbol{x}_{S^-(\boldsymbol{w}, i)})}[f(\boldsymbol{x}')], \tag{4}$$

$$\text{mono}- = \frac{1}{D-1} \sum_{i=0}^{D-2} \mathbb{I}_{|\delta_i^-| \leq |\delta_{i+1}^-|}. \tag{5}$$

Similar to faithfulness, we define a new variant of monotonicity by computing in the opposite order. Intuitively, each feature is ordered by the *inverse* of the weight vector, and then the fraction of indices $i$ such that the marginal improvement for feature $i$ is greater than the marginal improvement for feature $i+1$ is computed:

$$\delta_i^+ = \mathbb{E}_{\boldsymbol{x}' \sim \mathcal{D}(\boldsymbol{x}_{S^+(\boldsymbol{w}, i+1)})}[f(\boldsymbol{x}')] - \mathbb{E}_{\boldsymbol{x}' \sim \mathcal{D}(\boldsymbol{x}_{S^+(\boldsymbol{w}, i)})}[f(\boldsymbol{x}')], \tag{6}$$

$$\text{mono}- = \frac{1}{D-1} \sum_{i=0}^{D-2} \mathbb{I}_{|\delta_i^+| \leq |\delta_{i+1}^+|}. \tag{7}$$

The types of metrics discussed so far, faith and mono, each evaluate weight vectors by comparing an estimate of the marginal improvement of a set of features to their corresponding weights. Estimating the marginal improvement requires computing $f$ on different combinations of features, and it is possible that these combinations of features have very low density in $\mathcal{D}$, and are therefore unlikely to occur in $\mathcal{D}_{\text{train}}$. This is especially true for structured data or data where there are large low-density regions in $\mathcal{D}$ and may make the evaluations on $f$ unreliable. To help mitigate this issue, another paradigm of explainability evaluation metrics was proposed: remove-and-retrain (ROAR) [26]. In this paradigm, in order to evaluate the marginal improvement of sets of features, the model is retrained using a new dataset with the features removed. For example, rather than computing $\left|\mathbb{E}_{\boldsymbol{x}' \sim \mathcal{D}(\boldsymbol{x}_{F \setminus i})}[f(\boldsymbol{x}')] - f(\boldsymbol{x})\right|$,

we would compute $\left| f^*(\mathbb{E}_{\boldsymbol{x}' \sim \mathcal{D}(\boldsymbol{x}_{F \setminus i})}[\boldsymbol{x}']) - f(\boldsymbol{x}) \right|$, where $f^*$ denotes a model that has been trained on a modification of $\mathcal{D}_{\text{train}}$ where each datapoint has its $i$ features with highest weight removed. The original work advocated for reporting a curve of retrained model performance against number of features ablated [26]. In order to report a scalar metric, we propose four new metrics by combining the remove-and-retrain paradigm with faithfulness and monotonicity: **roar-faith+/-** and **roar-mono+/-**. That is, the definitions are similar to faith+/- and mono+/-, but $f$ is replaced with $f^*$ as defined above, accordingly. The formal definitions can be found in Appendix E.1. To compute a ROAR-based metric on all datapoints in the test set, the explainer must evaluate all datapoints in the training set to construct $D + 1$ ablated datasets, and then the model must be retrained for each of the datasets.

A caveat for all of the aforementioned metrics is that they evaluate each feature weight by computing the effect of removing the feature from a single set of features $S$. While this evaluation is sufficient for linear models, it may lead to unreliable measurements for nonlinear models such as neural networks. To address this, we use *Shapley values* [33, 16, 32, 48], which take into account the marginal improvement of a feature $i$ across *all possible* exponentially many sets with and without $i$. We consider two Shapley-based metrics: **shapley-mse** and **shapley-corr**, which involve computing the ground-truth Shapley values [33] for each feature, and then computing either the mean squared error (MSE) or Pearson correlation, respectively, between the weight vector and the set of ground-truth Shapley values. We give the formal definitions of the Shapley-based metrics in Appendix E. In Section 5, we also implement the infideltiy measure proposed by Yeh et al. [53]. This metric evaluates the important features of a datapoint by computing the difference in function values after significant perturbations on the input. Finally, note that Equation 1 defines "observational" conditional expectations [33, 1]. We also implement "interventional" conditional expectations [16, 49], which is defined by assuming the features in $S$ are independent of the remaining features. We discuss the tradeoffs between the two in the next section.

### 3.3  A guide to choosing metrics

Researchers may use any or all of the above metrics for evaluating and comparing different feature attribution explanation techniques. All metrics have strengths and weaknesses, and the most useful metric for each situation depend on the use-case, dataset, feature attribution technique, and computational constraints. Now we discuss the strengths and weaknesses of each metric type. The set of metrics can be naturally split into different dimensions: roar (retrain) versus non-roar, faith versus mono versus shapley, '+' versus '-' (alternatively for shapley, 'corr' versus 'mse'), and interventional versus observational conditional expectations.

For the first dimension, retraining the model with the most important features removed is especially important when the original model is not calibrated for out-of-distribution predictions [26], such as in high-dimensional applications like computer vision. However, retraining might fail to give an accurate evaluation in the presence of high feature correlations [37]. Furthermore, retraining the model incurs a much larger computational cost, which is especially important when a model is computationally intensive to train.

The second dimension is choosing between faith, mono, or shapley. Using a shapley-based metric will give a more accurate evaluation on explainability techniques that use shapley values. However, evaluating the shapley-based metrics require evaluating the model on a number of points that is exponential in the number of features. Therefore, the shapley-based metrics are extremely slow to evaluate on high-dimensional datasets. Faithfulness and monotonicity have far less computational constraints. The main difference between faithfulness and monotonicity is that faithfulness considers subsets of features by iteratively removing the most important features *with replacement*, while monotonicity does this *without replacement*. Therefore, the former is better for applications where the main question is which features would individually change the output of the model (and therefore may be better on datasets with less correlated features). The latter is better for applications where the main question is seeing the cumulative effect of adding features (and therefore performs comparatively better in the presence of correlated features).

Next, we discuss the pros and cons of using the '+' versus '−' options for faithfulness and monotonicity. Recall that the + means we order the features in increasing importance, while − means we order in decreasing importance. The + is more important in applications where discovering individual trends in the data is important, for example, drug discovery. The − is more important in

applications where we look at specific datapoints and the effect of changing individual features on the model prediction.

Finally, we discuss interventional versus observational conditional expectations. As pointed out in prior work, interventional conditional expectations are better for applications that require being 'true to the model', while observational conditional expectations are better for applications that require being 'true to the data', because observational conditional expectations tend to spread out importance among correlated features (even features that are not used by the model) [13]. For example, interventional conditional expectations are more appropriate in explaining why a model caused a loan to be denied, while observational conditional expectations are more appropriate in explaining the causal features in the drug response to RNA sequences [13]. Now we conclude with an example of an application for each of the components of the metrics.

- **roar** is useful for datasets with a high ratio of relevant features to datapoints (such as computer vision) when there are no computational constraints.
- **non-roar** is useful for datasets with a lower ratio of relevant features to datapoints (such as tabular data) or when there is not enough time to retrain the model.
- **faith** is useful on datasets with uncorrelated features, such as low-dimensional tabular data.
- **mono** is useful on datasets with correlated features, such as Census data that contains redundant information.
- **"+"** is useful for applications like drug response to RNA, where discovering individual trends in the data is important.
- **"−"** is useful for ablation-based tasks such as explaining the effect of removing individual features in a model that predicts loan repayment.
- **shapley-corr** is useful for datasets with a small number of highly correlated features.
- **shapley-mse** is useful when comparing Shapley-based explainers such as Kernel-SHAP, SHAPR, or other SHAP variants [48, 5, 25].
- **interventional conditional expectations** are useful for causal dataset tasks such as explaining the causal features in the drug response to RNA sequences.
- **observational conditional expectations** are useful for model-based tasks such as explaining the importance of each feature in a model that predicts loan repayment.

## 4  Synthetic Datasets

In this section, we describe the synthetic datasets used in our library. We start by discussing the benefits of synthetic datasets when evaluating feature attribution methods. Next, we describe our multivariate Gaussian and mixture of Gaussian datasets.

### 4.1  The case for synthetic data

As shown in Section 3.2, key to the metrics is computing the conditional expectation $\mathbb{E}_{\boldsymbol{x}' \sim \mathcal{D}(\boldsymbol{x}_S)}[f(\boldsymbol{x}')]$ for a subset $S$, datapoint $\boldsymbol{x}$, and trained model $f$. On real-world datasets, the conditional distribution $\mathcal{D}(\boldsymbol{x}_S)$ (defined in Equation 1) can only be approximated, and the approximation may be very poor when the conditional distribution defines a low-dimensional region of the feature space. Since all evaluation metrics require computing $\Theta(D)$ or $\Theta(2^D)$ such expectations, for each datapoint $\boldsymbol{x}$, is is likely that some evaluations will make use of a poor approximation. However, for the synthetic datasets that we define, the conditional distributions are known, allowing exact computation of the evaluation metrics.

Additionally, as we show in Section 5, synthetic datasets allow one to explicitly control all attributes of the dataset, which allows for targeted experiments, for example, investigating explainer performance as a function of feature correlation. For explainers such as SHAP [33] which assume feature independence, this type of experiment may be very beneficial. Finally, synthetic datasets can be used to simulate real datasets, which enables fair benchmarking of explainers with quantitative metrics.

### 4.2  Mutivariate Gaussian and mixture of Gaussians features

Now we describe the synthetic datasets in our library. In general, the datasets are expressed as $y = h(\boldsymbol{x})$, with y as label and $\boldsymbol{x}$ as feature vector. The generation is split into two parts, generating features $\boldsymbol{x}$, and defining a function to generate labels $y$ from $\boldsymbol{x}$.

We first describe feature generation, beginning with multivariate normal and mixture of Gaussians synthetic features. The multivariate normal distribution of a $D$-dimensional random vector $X = (X_1, ..., X_D)^T$ can be written as $X \sim \mathcal{N}(\boldsymbol{\mu}, \boldsymbol{\Sigma})$, where $\boldsymbol{\mu}$ is the $D$-dimensional mean vector, and $\boldsymbol{\Sigma}$ is the $D \times D$ covariance matrix. Without loss of generality, we can partition the $D$-dimensional vector $x$ as $X = (X_1, X_2)^T$. To compute the distribution of $X_1$ conditional on $X_2 = x_2^*$ where $x_2^*$ is a $K$-dimensional vector with $0 < K < D$, we can then partition $\boldsymbol{\mu}$ and $\boldsymbol{\Sigma}$ accordingly:

$$\boldsymbol{\mu} = \begin{bmatrix} \boldsymbol{\mu}_1 \\ \boldsymbol{\mu}_2 \end{bmatrix}, \quad \boldsymbol{\Sigma} = \begin{bmatrix} \boldsymbol{\Sigma}_{11} & \boldsymbol{\Sigma}_{12} \\ \boldsymbol{\Sigma}_{21} & \boldsymbol{\Sigma}_{22} \end{bmatrix}.$$

Then the conditional distribution is a new multivariate normal $(X_1 | X_2 = x_2^*) \sim \mathcal{N}(\boldsymbol{\mu}^*, \boldsymbol{\Sigma}^*)$ where

$$\boldsymbol{\mu}^* = \boldsymbol{\mu}_1 + \boldsymbol{\Sigma}_{12} \boldsymbol{\Sigma}_{22}^{-1}(x_2^* - \boldsymbol{\mu}_2), \quad \boldsymbol{\Sigma}^* = \boldsymbol{\Sigma}_{11} + \boldsymbol{\Sigma}_{12} \boldsymbol{\Sigma}_{22}^{-1} \boldsymbol{\Sigma}_{21}. \tag{8}$$

For any $x_2^* \in \mathbb{R}^K$, one can compute $\boldsymbol{\mu}^*$ and $\boldsymbol{\Sigma}^*$ and then generate samples from the conditional distribution. $\boldsymbol{\mu}$ can take any value, and $\boldsymbol{\Sigma}$ must be symmetric and positive definite. Similarly, we also include mixture of multivariate Gaussian features with derivation in Appendix D.

### 4.3 Labels

After creating a distribution of features by either a multivariate Gaussian or mixture of Gaussians, we then create the distribution of labels. The distributions we implement are `linear`, `piecewise constant`, and `nonlinear additive`.

Data labels are computed in two steps: *(1)* raw labels are computed from features, i.e. $y_{\text{raw}} = \sum_{n=1}^{D} \Psi_n(x_n)$ where $\Psi_n$ is a function that operates on feature $n$, and *(2)* final labels are normalized to have zero mean and unit variance. The normalization ensures that a baseline ML model, which always predicts the mean of the dataset, has an MSE of 1. This allows results derived from different types of datasets to be comparable at scale.

For `linear` datasets, $\Psi_n(x_n)$ are scalar weights, and we can rewrite the raw labels as $y_{raw} = \boldsymbol{w}^T \boldsymbol{x}$. For `piecewise constant` datasets, $\Psi_n(x_n)$ are piecewise constant functions made up of different threshold values (similar to Aas et al. [1]). For `nonlinear additive` datasets, $\Psi_n(x_n)$ are nonlinear functions including *absolute*, *cosine*, and *exponent* function adapted from Chen et al. [14]. Detailed specifications can be found in Appendix F.

## 5 Experiments

In this section, we describe our experiments in benchmarking several popular feature attribution methods across synthetic datasets.

### 5.1 Feature attribution methods

We compare six different feature attribution methods: SHAP [33], SHAPR [1], brute-force Kernel SHAP (BF-SHAP) [33], LIME [40], MAPLE [39], and L2X [14]. We also compare the methods to RANDOM explainer, which outputs random weights drawn from a standard normal distribution. See Appendix E for descriptions and implementation details for all methods. We report mean and standard deviation from three trials for all experiments.

### 5.2 Parameterized synthetic data experiments

Now we run experiments using multivariate Gaussian datasets described in Section 4. Without loss of generality, we can assume that the feature set is normalized (in other words, $\boldsymbol{\mu}$ is set to 0, and the diagonal of $\boldsymbol{\Sigma}$ is set to 1). In all sections except Section 5.3, we set the non-diagonal terms of $\boldsymbol{\Sigma}$ to $\boldsymbol{\rho}$, which allows for convenient parameterization of a global level of feature dependence and has been used in prior work [1].

We run experiments that compare six feature attribution methods on the eleven different evaluation metrics defined in Section 3.1 across several different datasets and ML models. In this section, we conduct experiments by varying one or two of these dimensions at a time while holding the other dimensions fixed (for example, we compare different datasets while keeping the ML model fixed) and in Appendix E.1, we give the exhaustive set of experiments. Throughout this section, we will

identify different types of failure modes, for example, failures for some explainability techniques over specific metrics (Table 1) or failures for some techniques on datasets with high levels of feature correlation (Figures 2 and 3).

**Performance across metrics** As shown in Table 1, the relative performance of explainers varies dramatically across metrics for a fixed multilayer perceptron trained on a `piecewise constant` dataset with $\rho = 0$. It is not surprising that SHAPR, which is an improvement of SHAP, performs well in Shapley metrics. In fact, SHAP, BF-SHAP, SHAPR, and LIME offer accurate approximation of ground truth Shapley values (>0.9 shapley-corr). In addition, LIME achieves top performance in three out of four ROAR-based metrics. Unexpectedly, none of the explainers outperformed random on mono-, suggesting that this metric is not helpful for this dataset and model. Another surprising observation is that while MAPLE performs well for faith+/-, and roar-mono+/-, it fails for roar-faith+/- by producing large negative scores, suggesting that it systematically ranks feature importance in an order opposite to the marginal improvement-based rankings in roar-faith+/-.

Table 1: Explainer performance across metrics. All performance numbers are from explaining a multilayer perceptron trained on the Gaussian piecewise constant dataset with $\rho = 0$.

| | RANDOM | SHAP | BF-SHAP | SHAPR | LIME | MAPLE | L2X |
|---|---|---|---|---|---|---|---|
| faith+($\uparrow$) | $-0.028_{\pm 0.022}$ | $\mathbf{0.922}_{\pm 0.020}$ | $0.921_{\pm 0.038}$ | $0.918_{\pm 0.039}$ | $0.859_{\pm 0.035}$ | $0.626_{\pm 0.050}$ | $-0.004_{\pm 0.100}$ |
| faith-($\uparrow$) | $-0.022_{\pm 0.023}$ | $0.970_{\pm 0.006}$ | $\mathbf{0.979}_{\pm 0.005}$ | $0.977_{\pm 0.004}$ | $0.918_{\pm 0.010}$ | $0.647_{\pm 0.045}$ | $0.002_{\pm 0.080}$ |
| mono+($\uparrow$) | $0.538_{\pm 0.012}$ | $\mathbf{0.720}_{\pm 0.018}$ | $0.708_{\pm 0.012}$ | $0.719_{\pm 0.019}$ | $0.667_{\pm 0.032}$ | $0.712_{\pm 0.008}$ | $0.562_{\pm 0.024}$ |
| mono-($\uparrow$) | $\mathbf{0.467}_{\pm 0.006}$ | $0.433_{\pm 0.019}$ | $0.407_{\pm 0.016}$ | $0.435_{\pm 0.012}$ | $0.428_{\pm 0.014}$ | $0.440_{\pm 0.017}$ | $0.430_{\pm 0.040}$ |
| roar-faith+($\uparrow$) | $0.003_{\pm 0.028}$ | $0.461_{\pm 0.095}$ | $0.502_{\pm 0.038}$ | $0.468_{\pm 0.082}$ | $\mathbf{0.585}_{\pm 0.046}$ | $-0.429_{\pm 0.018}$ | $0.045_{\pm 0.060}$ |
| roar-faith-($\uparrow$) | $0.008_{\pm 0.049}$ | $0.581_{\pm 0.024}$ | $0.523_{\pm 0.006}$ | $0.559_{\pm 0.026}$ | $\mathbf{0.621}_{\pm 0.019}$ | $-0.339_{\pm 0.013}$ | $0.052_{\pm 0.038}$ |
| roar-mono+($\uparrow$) | $0.474_{\pm 0.016}$ | $0.747_{\pm 0.028}$ | $0.734_{\pm 0.011}$ | $0.730_{\pm 0.022}$ | $0.707_{\pm 0.024}$ | $0.425_{\pm 0.009}$ | $0.500_{\pm 0.027}$ |
| roar-mono-($\uparrow$) | $0.492_{\pm 0.019}$ | $0.721_{\pm 0.032}$ | $0.709_{\pm 0.04}$ | $0.713_{\pm 0.044}$ | $\mathbf{0.745}_{\pm 0.020}$ | $0.471_{\pm 0.016}$ | $0.451_{\pm 0.041}$ |
| shapley-corr($\uparrow$) | $0.001_{\pm 0.014}$ | $0.992_{\pm 0.005}$ | $\mathbf{1.000}_{\pm 0.001}$ | $0.998_{\pm 0.001}$ | $0.955_{\pm 0.009}$ | $0.735_{\pm 0.038}$ | $0.073_{\pm 0.084}$ |
| shapley-mse($\downarrow$) | $1.134_{\pm 0.040}$ | $0.003_{\pm 0.001}$ | $\mathbf{0.000}_{\pm 0.000}$ | $\mathbf{0.000}_{\pm 0.000}$ | $0.026_{\pm 0.001}$ | $0.071_{\pm 0.007}$ | $0.188_{\pm 0.022}$ |
| infidelity($\downarrow$) | $0.169_{\pm 0.014}$ | $0.044_{\pm 0.009}$ | $0.048_{\pm 0.01}$ | $\mathbf{0.043}_{\pm 0.008}$ | $0.049_{\pm 0.012}$ | $0.073_{\pm 0.007}$ | $0.122_{\pm 0.021}$ |

**Performance across dataset types and feature correlations** Next, we explore how the type of dataset and feature correlation affects performance of explainers on a decision tree model with the faithfulness metric. As shown in Figure 2, a general trend is that explainers become less faithful as feature correlation increases. Explainers such as SHAP assume feature independence and tend to perform well when features are indeed independent ($\rho = 0$). This is especially apparent with the `linear` dataset, where all performance of most methods cluster around 0.9 at $\rho = 0$. However, LIME's performance drops as much as $\sim 90\%$ when features are almost perfectly correlated ($\rho = 0.99$). On the other hand, for both the `nonlinear additive` and `piecewise constant` datasets, MAPLE's performance stayed relative stable across values of $\rho$.

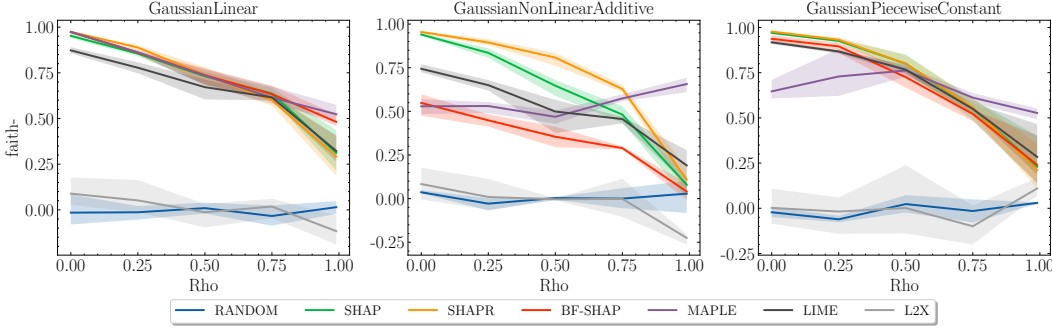

Figure 2: Results for faith- on a multilayer perceptron trained on three types of Gaussian datasets.

**Performance across ML models** Next, we train three ML models: linear regression, decision tree, and multilayer perceptron, with a `piecewise constant` dataset and compare faith-. Figure 3 shows that as in Figure 2, explainer performance drops as features become more correlated. Most explainers perform well for linear regression up to $\rho = 0.75$. The performance of SHAP, SHAPR, and LIME remain relatively consistent across ML models. In contrast, BF-SHAP performs significantly worse

on the tree model. The nearly consistent negative faith- score of MAPLE on the tree model provides additional evidence to Table 1 that in some cases, the most important feature weights MAPLE predicts tend be the least important.

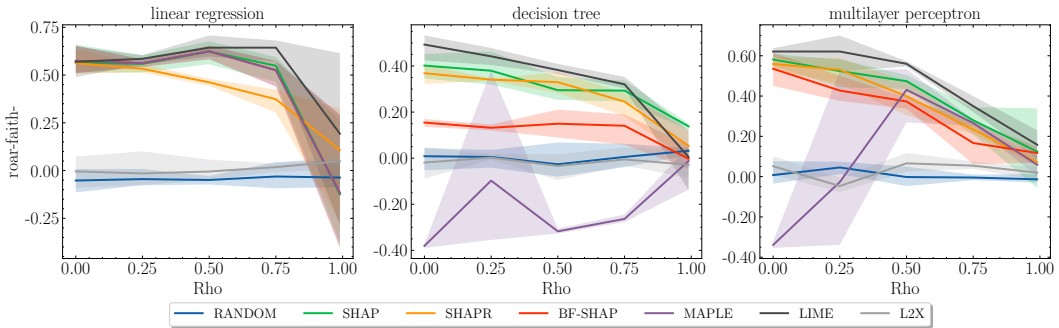

Figure 3: Results for faith- for three types of ML models– linear regression, decision tree, and multilayer perceptron– trained on a Gaussian piecewise constant dataset.

## 5.3 Simulating the wine dataset

In this section, we demonstrate the power and flexibility of synthetic datasets by simulating the popular wine dataset [15, 47] with synthetic features so that it can be used to efficiently benchmark feature attribution methods.

The white wine dataset has 11 continuous features ($x_{\text{real}}$) and one integer quality rating ($y_{\text{real}}$) between 0 and 10. In this section, it is formulated as a regression task, but it can also be formulated as a multi-class classification task. The features are first normalized to have zero mean and unit variance, then an empirical covariance matrix is computed (Appendix Figure 5), which is then used as the input covariance matrix to generate synthetic multivariate Gaussian features ($x_{\text{sim}}$). Simulated wine quality ($y_{\text{sim}}$) is labeled by a $k$-nearest neighbor model based on real datapoints ($x_{\text{real}}, y_{\text{real}}$).

We evaluate how close the simulated dataset is to the real one in two steps. First, we compute the Jensen-Shannon Divergence (JSD) [51] of the real and synthetic wine datasets. JSD measures the similarity between two distributions, it is bounded between 0 and 1, and lower JSD suggests higher similarity between two distributions. The JSD of marginal distributions between the real empirical features and the synthetic Gaussian features has a mean of $0.20$, and the JSD of real and synthetic targets is $0.23$, suggesting a good fit. Second, we train three types of ML models on both simulated and real wine datasets and compare the MSE of explanations on a common held-out real test set. As shown in Appendix Table 4, consistent low MSE across ML models and explainers suggest that the simulated dataset is a good proxy for the original wine dataset for evaluating explainers.

Next, we compute evaluation metrics for five different explainers on the synthetic wine dataset. Note that computing these metrics accurately is not possible on the real wine dataset, as the conditional distribution is unknown. As shown in Table 2, SHAP performs well on the Shapley metrics, consistent with Table 1. Both LIME and MAPLE outperform SHAP on faith+. MAPLE achieves top performance on mono-, however, none of the explainers significantly outperform RANDOM.

Table 2: Explainer performance on the simulated wine dataset across metrics. All performance numbers are from explainers explaining a decision tree model.

|  | RANDOM | SHAP | SHAPR | LIME | MAPLE | L2X |
|---|---|---|---|---|---|---|
| faith- (↑) | $0.012_{\pm 0.011}$ | $0.461_{\pm 0.034}$ | $\mathbf{0.528}_{\pm 0.032}$ | $0.237_{\pm 0.031}$ | $-0.007_{\pm 0.036}$ | $-0.010_{\pm 0.032}$ |
| faith+ (↑) | $0.025_{\pm 0.038}$ | $0.488_{\pm 0.023}$ | $0.579_{\pm 0.015}$ | $\mathbf{0.595}_{\pm 0.022}$ | $0.556_{\pm 0.021}$ | $0.055_{\pm 0.035}$ |
| mono- (↑) | $0.490_{\pm 0.004}$ | $0.502_{\pm 0.010}$ | $\mathbf{0.518}_{\pm 0.005}$ | $0.500_{\pm 0.013}$ | $0.506_{\pm 0.011}$ | $0.492_{\pm 0.001}$ |
| mono+ (↑) | $0.523_{\pm 0.010}$ | $\mathbf{0.556}_{\pm 0.012}$ | $0.551_{\pm 0.009}$ | $0.539_{\pm 0.005}$ | $0.513_{\pm 0.008}$ | $0.522_{\pm 0.008}$ |
| shapley-corr (↑) | $0.011_{\pm 0.027}$ | $0.815_{\pm 0.024}$ | $\mathbf{0.945}_{\pm 0.002}$ | $0.692_{\pm 0.019}$ | $0.669_{\pm 0.007}$ | $0.035_{\pm 0.055}$ |
| shapley-mse (↓) | $1.032_{\pm 0.022}$ | $0.014_{\pm 0.003}$ | $\mathbf{0.004}_{\pm 0.001}$ | $0.032_{\pm 0.005}$ | $0.041_{\pm 0.001}$ | $0.055_{\pm 0.001}$ |
| infidelity (↓) | $0.224_{\pm 0.137}$ | $0.13_{\pm 0.113}$ | $0.212_{\pm 0.146}$ | $\mathbf{0.129}_{\pm 0.111}$ | $0.129_{\pm 0.114}$ | $0.203_{\pm 0.109}$ |

### 5.4 Recommended usage

Throughout Section 5, we gave a sample of the types of experiments that can be done using XAI-BENCH (recall that our comprehensive experiments are in Appendix E.1). For researchers looking to develop new explainability techniques, we recommend benchmarking new algorithms across all metrics using our synthetic multivariate Gaussian and mixture of Gaussian datasets with different values of $\rho$. These datasets give a good initial picture of the efficacy of new techniques. For researchers with a dataset and application in mind, we recommend converting the dataset into a synthetic dataset using the technique described in Section 5.3. Note that converting to a synthetic dataset also gives the ability to evaluate explainability techniques on perturbations of the original covariance matrix, to simulate robustness to distribution shift. Finally, researchers can decide on the evaluation metric that is most suitable to the application at hand. See Section 3.3 for a guide to choosing the best metric based on the application.

## 6 Societal Impact

Machine learning models are more prevalent now than ever before. With the widespread deployment of models in applications that impact human lives, explainability is becoming increasingly more important for the purposes of debugging, legal obligations, and mitigating bias [33, 55, 7, 18]. Given the importance of high-quality explanations, it is essential that the explainability methods are reliable across all types of datasets. Our work seeks to speed up the development of explainability methods, with a focus on catching edge cases and failure modes, to ensure that new explainability methods are robust before they are used in the real world. Of particular importance are improving the reliability of explainability methods intended to recognize biased predictions, for example, ensuring that the features used to predict criminal recidivism are not based on race or gender [30]. Frameworks for evaluating and comparing explainability methods are an important part of creating inclusive and unbiased technology. As pointed out in prior work [17], while methods for explainability or debiasing are important, they must be part of a larger, socially contextualized project to examine the ethical considerations of the machine learning application.

## 7 Conclusions and Limitations

In this work, we released a set of synthetic datasets along with a library for benchmarking feature attribution algorithms. The use of synthetic datasets with known ground-truth distributions makes it possible to exactly compute the conditional distribution over any set of features, enabling computations of several explainability evaluation metrics, including ground-truth Shapley values, ROAR, faithfulness, and monotonicity. Our synthetic datasets offer a variety of parameters which can be configured to simulate real-world data and have the potential to identify failure modes of explainability techniques, for example, techniques whose performance has a negative correlation with dataset feature correlation. We showcase the power of our library by benchmarking several popular explainers with respect to ten evaluation metrics across a variety of settings.

Furthermore, despite the fact that the synthetic datasets aim to cover a broad range of feature distributions, correlations, scales, and target generation functions, there is almost certainly a gap between synthetic and real-world datasets. However, as discussed before, it is often the case that we do not know the ground truth generative model of real datasets, thus making it impossible to compute many objective metrics. Hence, there is a trade-off between data realism and ground truth availability.

Note that our library is **not** meant to be a replacement for human interpretability studies. Since the goals of explainability methods are inherently human-centric, the only foolproof method of evaluating explanation methods are to use human trials. Rather, our library is meant to substantially speed up the process of development, refinement, and identifying failures, before reaching human trials.

Overall, we recommend developing new explainability methods in this library, and then conducting human trials on real data. Our library is designed to substantially accelerate the time it takes to move new explainability algorithms from development to deployment. With the release of API documentation, walkthroughs, and a contribution guide, we hope that the scope of our library can increase over time.

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
