# OpenReview forum: "Synthetic Benchmarks for Scientific Research in Explainable Machine Learning"
_NeurIPS.cc/2021/Track/Datasets_and_Benchmarks/Round1 — Submitted to NeurIPS 2021 Datasets and Benchmarks Track (Round 1)_

### Official Review · Reviewer_RWjA · 2021-07-03
**An Interesting Benchmarking Approach, but too Limited in Scope at this Point**

**Rating:** 5
**Confidence:** 5

**Strengths:**

-	It is important to evaluate the quality of ML explanations, and to date there exists little no such work, where this is done systematically. Using synthetic data sets to evaluate various explanations techniques with respect to ground-truth explanations, constitutes a first step into this direction.
-	The package provides several established evaluation measures which should help compare different explanation methods in the future.


**Weaknesses:**

Failure Modes
- It is a bit unclear what the purpose of this work is: In the abstract, introduction, and conclusion you emphasize that one of the major purposes of this work lies within identifying the failure modes of popular explanation techniques. If this is the major purpose of the benchmark, then it is strange that you never define what a ‘failure mode’ entails. At the same time, there is already a very strong understanding of some of these method’s failure modes (e.g., see [1] and the downstream citations), and drawing a connection to this line of work would certainly be helpful, if this was the primary motivation of your benchmark.
- The benchmark is limited to 5 explanation techniques. In fact, for these explanation techniques there already exists a plethora of works that show the failure modes of these methods: e.g., [1, 2, 3].
- There is little discussion on the hyperparameters chosen for the experimental evaluation, while it has already been established in the literature that the explanations substantially vary depending on the choice of hyperparameters (e.g., [2, 3]). A benchmark that does not take this into consideration is difficult to be taken seriously.

Explanation Models
- The benchmark is limited to 5 explanation models. For these 5 models, the hyperparameter choices are not discussed.

Data sets
- The benchmark is limited to Gaussian data sets. For the purpose of getting closer to practical use-cases, it would be fruitful to also consider a mixture of both Gaussian and Discrete data distributions (Bernoulli, Poisson, Categorical).

Measures
- It would be good to integrate the infidelity (i.e., faithfulness) measure suggested in [4] as it provides an objective faithfulness measure.

[1] ‘Explanations can be manipulated and geometry is to blame’, Dombrowski et al (NeurIPS 2019)
[2] ‘SAM: the sensitivity of attribution methods to hyperparameters’, Bansal et al (CVPR 2020)
[3] ‘Explaining the explainer: a first theoretical analysis of lime’, Garreau and von Luxburg (AISTATS 2020)
[4] ‘On the (In)fidelity and Sensitivity of Explanations’, Yeh et al 2019 (NeurIPS)


**Additional Feedback:**

- It would have been useful to enable the line numbering.
- Table 2: the caption says that you are considering MLPs; the text says: 'As shown in Table 2, the [...] for a fixed decision tree model trained on the piecewise [...].' So, which model is unerlying the results in Table 2 - a tree model or a MLP model?
- Why does Brute-Force SHAP has no correlation of 1 with the shapley values? This is unintuitive, and should warrant some explanations.
- Why are results for SHAPR not shown in Table 3?

**Clarity:**

- The paper is written in a clear way. However, from my point of view, it is unclear what the purpose of the author's package is at this point: please refer to the 'Weaknesses' section to see more detailed comments.

- After reading the results of your paper, I have been wondering why practioners and researchers should keep interacting with this package, and not simply go and use the individual libraries?

**Correctness:**

The evaluation seems to be performed correctly. Again, it would be good to integrate the infidelity (i.e., faithfulness) measure suggested in [4].

**Documentation:**

The author's github package provides sufficient support to reproduce the results.

**Ethics:**

There are no ethical concerns.

**Relation To Prior Work:**

The paper misses some of the relevant works. Please refer to the 'Weaknesses' section to see more detailed comments.

**Summary And Contributions:**

The authors provide a benchmarking library for 5 popular black-box ML explanation techniques: their library includes synthetic data sets, evaluation measures, and 5 different explanation techniques (the code for the explanation models is mostly loaded from already existing libraries). Using their library, the authors then conduct a benchmark on their generated data sets, mostly using already existing evaluation measures (faithfulness, monotonicity, ROAR, etc.).

---

> ### Author Response · Authors · 2021-07-09
> **We agree: we will add hyperparameter options, more dataset distribution options, and more discussion and clarification on failure modes**
>
> We thank the reviewer for their detailed comments and suggestions. We reply to each point below.
>
> 1. Hyperparameters. Thank you for the suggestion! We think this will significantly improve the usefulness of our library. For each explainability algorithm, we will identify several hyperparameters based on the papers cited by the reviewer, and add them as configurable options in our config files. We will complete this and update the paper by the end of the author response period (July 14th).
> 2. Datasets. We agree that adding more data distribution options will improve our work. We are now adding discrete distributions (e.g. Bernoulli) as the reviewer suggested.
> 3. Algorithms and metrics. We are planning to add more explainability algorithms, such as breakDown [1], and we will add the infidelity metric as the reviewer requested. If the reviewer has any more suggestions, please let us know.
> 4. Failure modes: we agree with the reviewer’s thoughts on failure modes. We will (a) cite all of the relevant work suggested by the reviewer, (b) flesh out the initial discussion started in Sections 5.2 and 5.4, including a specific definition of “failure mode”, and (c) since we agree that this is not the *main* contribution of our work, we will balance the emphasis of our other contributions in the abstract, introduction, and conclusion.
> 5. “Additional feedback” suggestions. We thank the reviewer for spotting these inconsistencies, which we will fix.
>
> The review mentioned that there is little to no work in systematically evaluating explainability techniques, and our synthetic benchmarks are a first step in this direction. We are committed to making our repository adopted by the community, which is why we will add your suggestions (more hyperparameter options, data distributions, metrics, and explainability algorithms). Finally, we respectfully ask that the reviewer considers increasing their score if they find our response satisfactory. If it is still unsatisfactory, please let us know if there is anything else that we can do to make our repository more useful to the explainability community.
>
> [1] https://arxiv.org/abs/1804.01955

---

> > ### Comment · Reviewer_RWjA · 2021-07-20
> > **Final Comments**
> >
> > I very much appreciate the authors' effort in addressing all my initial concerns. I adjusted my score from 4 to 5. While I think that the benchmark is useful, I still believe that its current scope is limited to few very popular methods which are well understood by the literature. It would perhaps be of greater use to the community to integrate and compare less commonly used (e.g., rule-based) methods (e.g., [1]), or explore the connections to counterfactual explanations systematically (e.g., [2, 3, 4]).
> >
> > - [1]: Faithful and Customizable Explanations of Black Box Models, Lakkaraju et al 2019, AIES
> > - [2]: Towards Unifying Feature Attribution and Counterfactual Explanations: Different Means to the Same End, Mothilal et al (2020), arXiv:2011.04917
> > - [3]: Learning Counterfactual Explanations for Tabular Data, Pawelczyk et al (2020), WWW
> > - [4]: Actionable Recourse in Linear Classification, Ustun et al (2019), FAT*

---

### Official Review · Reviewer_4JiN · 2021-07-04
**Well motivated but highly unclear why the proposed metrics will realistically help improve the current caveats of explanation methods and lack of metrics.s.**

**Rating:** 5
**Confidence:** 4
**Correctness:** Please see above concerns re- challen…

**Strengths:**

1. The paper is well written
2. I agree with the motivation of using synthetic datasets and the method using which synthetic datasets are created to be able to measure the proposed metrics in the most unbiased way possible.
3. The authors have tried to exhaustively compare existing methods under the benchmark and proposed metrics.

**Weaknesses:**

1. The proposed suite of evaluation metrics is primarily motivated by looking at perturbations of features and their impact on model performance. The authors make choices of what perturbations are useful measure of such performance change (e.g. conditional distribution). I believe the motivation for pushing for such a metric is actually very weak. The authors do not provide intuition about why this is actually a good measure of faithfulness and why the choice of conditional distribution a reasonable one.

2. While I understand that there is precedent to do this from prior literature, such metrics have also practically been proposed as *explanation* methods themselves (think feature occlusion and perturbation methods). Reconciling this is challenging but authors proposing a benchmark should I think really be making a convincing argument why this is a good idea. I urge the authors to justify their metrics a bit beyond based on prior work.

3. The metric based on retraining the model is even more confusing (i.e. roar-*). If the model changed (because we retrained), what black-box are we really explaining attributions for?

4. Shapley values have been argued to have significant challenges when used as attributions - https://arxiv.org/abs/2002.11097 at least as far as ML applications are concerned. In this case then, the motivation for relying on shapley as a metric seems unjustified to me. If there are criticisms of a metric, it is important to consider how they can be improved upon or justified more in my opinion. If the authors really are convinced shapley based metrics are still reasonable, please justify the choices more than what is currently in the draft.

**Additional Feedback:**

Please see my comments on weaknesses.

**Clarity:**

Yes I believe the paper is well written and the motivation is clear. My main concerns are with the execution and choices made along the way.

**Documentation:**

I have several technical concerns with the framing of the benchmark and hence I believe more work needs to be done before end-users of explanations can start using the metrics to truly test the utility of their explanations/feature attributions.

**Ethics:**

The ethical concerns in my opinion will arise if the current set of metrics are used to compare explanation methods as I don't believe they are clearly well motivated. If the current technical challenges are fixed, I don't believe there are major ethical concerns modulo all caveats that come with using explanations themselves (which are mostly not exclusive to such a benchmark in any way).

**Relation To Prior Work:**

Prior work is well contextualized but I believe authors are relying too much on prior precedent to justify their metrics rather than justifying their utility technically.

**Summary And Contributions:**

The authors propose a synthetic dataset + evaluation metrics to benchmark feature attribution methods in explanations. The authors propose versions of faithfulness and monotonicity metrics motivated by versions of similar metrics proposed in existing explanability literature. Most of the motivated metrics rely on knowing the conditional/marginal distributions of the data. In order to remove estimation challenges, authors create synthetic data that will allow to measure the metrics with known and true conditional/marginal distributions. Several existing feature attribution methods are compared using the proposed metrics on synthetic data for comparison.

---

> ### Author Response · Authors · 2021-07-09
> **We completely agree: we will add more metrics and write a detailed discussion of the strengths, weaknesses, and example applications for each metric**
>
> We thank the reviewer for the valuable comments that will improve our paper. We agree with the reviewer about the downsides of some of the metrics we used. All empirical metrics have certain strengths and weaknesses, and so we will make the following updates: (a) add more options for existing metrics (e.g., different methods for computing conditional expectations), (b) add new metrics, and (c) write a thorough discussion of the strengths, weaknesses, and example applications for all of the metrics that we implemented. One of the strengths of our codebase is that new metrics can be added easily. We respectfully ask that the reviewer consider increasing their score if they find our reply satisfactory.
>
> We give details for each specific point below.
>
> - (1&2) The reviewer is correct that the computation of conditional expectations for explanations is disputed in the research community. For example, [1] (which Reviewer kV5h mentioned as well) points out that some work uses “observational” expectations [2,3] and other work uses “interventional” expectations [4,5], and the best choice depends on the application. Therefore, we will keep our current version of conditional expectation and also add the “interventional” version, and we will add a discussion of the strengths and weaknesses of each approach.
> - (3) Similarly, for the ROAR metric, some work explains its strengths [6,7], while other work points out its flaws [8,9]. The usefulness of ROAR-based metrics likely depends on the specific application (for example, whether the model is accurate in low-density regions of the feature space).
> - (4) Once again, the reviewer is correct that the benefits of Shapley values are debated [10,11] even though it is one of the most widely used techniques in explainable AI.
>
> Overall, there are many debates about the best metrics to use in the current explainability literature, and there is no single best metric—the usefulness of each metric depends on the application (including the model, dataset, explainability technique, and goals of the users). One of the strengths of our repository is that users can choose among 10+ different metrics, picking the best one for their application. We will also include a guide to picking the best metric.
>
> If the reviewer has any more questions, or any remaining weaknesses which would cause you to consider increasing your score if addressed, please let us know.
>
> [1] https://arxiv.org/abs/2006.16234
> [2] https://arxiv.org/abs/1705.07874
> [3] https://arxiv.org/abs/1903.10464
> [4] https://www.andrew.cmu.edu/user/danupam/datta-sen-zick-oakland16.pdf
> [5] https://arxiv.org/abs/1908.08474
> [6] https://arxiv.org/abs/1806.10758
> [7] https://arxiv.org/abs/2102.06761
> [8] https://arxiv.org/abs/2007.07584
> [9] https://arxiv.org/abs/1905.00780
> [10] https://arxiv.org/abs/1908.08474
> [11] https://arxiv.org/abs/1910.13413

---

> > ### Comment · Reviewer_4JiN · 2021-07-21
> > **Thank you for addressing my concerns**
> >
> > Authors have made considerable effort in addressing the concerns I raised, I will therefore increase my score to 5. I do think authors should strongly reconsider the framing of shapley value based metrics, instead of merely highlighting limitations. I am not being unnecessarily harsh but a benchmark needs to truly be carefully thought through if it has to become a standard for the community to use. Its a laudable effort, and I genuinely encourage the authors to continue to polish this benchmark.

---

### Official Review · Reviewer_kV5h · 2021-07-04
**Helpful contribution to literature on evaluating explanations**

**Rating:** 7
**Confidence:** 4
**Correctness:** To my knowledge, they are correct.

**Strengths:**

There is still relatively little literature on evaluating explanations on common benchmarks and comparing their performances under different conditions. The systematic comparison in this paper is a good start that many others can build on. The results of their systematic comparison are intriguing and are a good source for further investigation, e.g. on why most explainers perform badly on the mono- measure, and why some explainers are better at handling correlations than others.

The method for generating synthetic datasets is also important for the general project of evaluating explanations.

The authors are very clear about the limitations of their method relative to human studies, which is a refreshing attitude in ML.

**Weaknesses:**

There's an ongoing debate on what is the right kind of conditional expectation to use (interventional or observational; see e.g. https://arxiv.org/pdf/2006.16234.pdf ). Methods like SHAP are able to use either kind in their estimates. It would be good for the authors to make clear which kind of conditional expectation they are using in their evaluations, and possibly even do a comparison for how SHAP performs on their metrics if you use the other kind of conditional expectation.



**Additional Feedback:**

Possibly worth citing: a very recent paper that also describes a library to evaluate locally linear explanations on a few metrics: https://arxiv.org/abs/2106.00461

The notation used to describe the metrics in Section 3.2 is unwieldy and takes a lot more effort to understand compared to other papers describing similar metrics (e.g. faithfulness). It would be good to have more conceptual summaries of what's going on in the equations, or to tweak the notation so that there are fewer complicated subscripts to process.

Given that the synthetic datasets are motivated by wanting 'ground truth' conditional expectations, it would have been interesting to see how the evaluation metrics as computed using 'ground truth' conditional expectation differs from the metrics as computed using SHAP's methods of estimating conditional expectations. The results would shed light on a lot of the ongoing debates about SHAP's pros and cons.

**Clarity:**

Yes, the paper is very clear on the intent, the scope of the paper, and the limitations.

**Documentation:**

Yes

**Relation To Prior Work:**

Yes

**Summary And Contributions:**

 - Provides a tool to evaluate different ML explanations on a few metrics
 - Provides a method for generating synthetic datasets with known conditional distributions that resemble real datasets, which allows better evaluation of explanations
 - Their experiments provide some preliminary data on which explainers perform best on which metrics, and how explainer performance is affected by correlations in the data

---

> ### Author Response · Authors · 2021-07-09
> **Thanks for the suggestions; we will add interventional conditional expectations**
>
>
> We thank the reviewer for their favorable review and insightful comments. The reviewer mentioned improvements in the “weaknesses” and “additional feedback” sections, all of which we will be able to incorporate to make our work even stronger. Specifically,
> 1. In our paper, we used observational conditional expectations following some prior work. We agree with the main idea from the paper that the reviewer shared, (observational vs. interventional is application-dependent), and we will add interventional conditional expectations as an option to all of the explainers and metrics that we already implemented (note that interventional is simpler to implement compared to observational, which we already did). Thank you for the suggestion! We think this will significantly improve the usefulness of our library.
> 2. We thank the reviewer for sharing the very recent related paper. We will add it to the related work section.
> 3. We agree that we should add much more intuition behind the metric equations. We will add this to the paper.
> 4. We agree that it would be interesting to compare the “observational” conditional expectation to the SHAP approximation of conditional expectation, and we will add this functionality to our repository.

---

### Author Response · Authors · 2021-07-14
**Revised paper following reviewers' comments**

Dear reviewers and AC, we have now addressed all of the suggestions and concerns mentioned by the reviewers.

The reviews mainly focused on (i) new feature requests, or (ii) concerns about evaluation metrics that we had implemented.
For (i), we were able to address all of the features requested (listed in detail below).
For (ii), we added more metric options and included a list of the strengths, weaknesses, and example use-cases for each metric (Section 3.3).

We thank the reviewers once again for the helpful suggestions, which we believe have substantially improved our paper. All reviewers expressed the idea that current explainability literature is lacking thorough comparisons, and our paper makes a step in that direction. We are committed to making our repository adopted by the community, so please let us know if there is anything else that we can do to make it more useful. We give a list of the changes below.

- We added a new explainer: [breakDown](https://github.com/abacusai/xai-bench/blob/main/custom_explainers/breakdown.py) [1].
- We added a new metric: [infidelity](https://github.com/abacusai/xai-bench/blob/main/custom_metrics/infidelity.py) [2].
- We added interventional conditional expectations ([here](https://github.com/abacusai/xai-bench/blob/2fe31c1d89b9284abc72112b69b1686e6e1b31e7/custom_metrics/faithfulness.py#L14) is an example).
- We added hyperparameter options for SHAPR, LIME, MAPLE, and L2X to the [config file](https://github.com/abacusai/xai-bench/blob/main/configs/experiment_config.jsonc).
- We added a new synthetic dataset: [multinomial](https://github.com/abacusai/xai-bench/blob/main/synthetic_datasets/synthetic_multinomial.py).
- We updated the paper:
    - We added more intuition for the metric equations (Section 3.2).
    - We added a list of strengths, weaknesses, and example use cases for the metrics (Section 3.3).
    - We clarified our repository’s capabilities with respect to identifying failure modes as follows. One of the strengths of our repository is that it is easy to identify certain failures such as explainers which fail for some metrics on datasets with high feature correlation. However, other work has defined and identified other types of failure modes, for example, explainers which fail due to adversarial perturbations [3,4] or small changes in hyperparameters [5,6]. To avoid confusion, in our paper we removed the phrase identify failure mode from the abstract, introduction, and conclusion, and we define what we mean by failure modes more precisely in Section 5.2.
    - We added new experiments with the infidelity evaluation metric.
    - We fixed all of the other smaller inconsistencies noticed by the reviewers.

Thanks again to all reviewers.

[1] https://arxiv.org/abs/1804.01955
[2] https://arxiv.org/abs/1901.09392
[3] https://arxiv.org/abs/1906.07983
[4] https://arxiv.org/abs/1911.02508
[5] https://arxiv.org/abs/2003.08754
[6] https://arxiv.org/abs/2001.03447

---

### Decision · Program_Chairs · 2021-07-26

**Decision:**

Reject

**Comment:**

The main concern is that the set of metrics proposed by the benchmark are not well justified, and justification is especially crucial for synthetic datasets.  The authors are advised to more carefully consider which metrics they recommend (perhaps focusing on fewer, more reliable metrics) and to include a deeper discussion of the limitations of the proposed metrics.